# Linewidth Measurement of a Narrow-Linewidth Laser: Principles, Methods, and Systems

**DOI:** 10.3390/s24113656

**Published:** 2024-06-05

**Authors:** Jia-Qi Chen, Chao Chen, Jing-Jing Sun, Jian-Wei Zhang, Zhao-Hui Liu, Li Qin, Yong-Qiang Ning, Li-Jun Wang

**Affiliations:** 1State Key Laboratory of Luminescence Science and Technology, Changchun Institute of Optics, Fine Mechanics and Physics, Chinese Academy of Sciences, Changchun 130033, China; chenjq018@163.com (J.-Q.C.); sunjingjing21@mails.ucas.ac.cn (J.-J.S.); zjw1985@ciomp.ac.cn (J.-W.Z.); liuzhaohui22@mails.ucas.ac.cn (Z.-H.L.); qinl@ciomp.ac.cn (L.Q.); ningyq@ciomp.ac.cn (Y.-Q.N.); wanglj@ciomp.ac.cn (L.-J.W.); 2University of Chinese Academy of Sciences, Beijing 100049, China; 3Xiongan Innovation Institute, Chinese Academy of Sciences, Xiongan 071800, China

**Keywords:** narrow-linewidth laser, delayed self-heterodyne, Brillouin, linewidth

## Abstract

Narrow-linewidth lasers mainly depend on the development of advanced laser linewidth measurement methods for related technological progress as key devices in satellite laser communications, precision measurements, ultra-high-speed optical communications, and other fields. This manuscript provides a theoretical analysis of linewidth characterization methods based on the beat frequency power spectrum and laser phase noise calculations, and elaborates on existing research of measurement technologies. In addition, to address the technical challenges of complex measurement systems that commonly rely on long optical fibers and significant phase noise jitter in the existing research, a short-delay self-heterodyne method based on coherent envelope spectrum demodulation was discussed in depth to reduce the phase jitter caused by 1/*f* noise. We assessed the performance parameters and testing conditions of different lasers, as well as the corresponding linewidth characterization methods, and analyzed the measurement accuracy and error sources of various methods.

## 1. Introduction

The research and application of advanced laser technology have been highly valued with the rapid development of optical communications, quantum technology, artificial intelligence, and other fields. Narrow-linewidth lasers have become key devices in fields such as long-distance optical space communications, ultra-high-speed fiber-optic communications, high-sensitivity optical sensing, and quantum precision measurement [1,2,3,4] due to their high monochromaticity, high coherence, low noise, high stability, and dynamic single-mode characteristics. In satellite laser communications, the longest link between the satellite and the ground is approximately 80,000 km, and the coherent laser transmission rate between satellites can reach 10 Gb/s. An ultra-narrow spectral linewidth and extremely high coherence based on narrow-linewidth lasers can achieve ultra-high-speed and large-capacity data transmission in space optical communications [5,6]. In terms of precision measurement, such as for atomic clocks, atomic magnetometers and LiDAR, laser heterodyne interferometry measurement systems have more stringent performance requirements for lasers [7,8,9,10]. A narrow-linewidth laser characterizes physical quantities such as the time reference, gravity, and magnetic field by pumping atoms and manipulating quantum states [11,12,13,14]. In the field of ultra-high-speed optical communication, different coherent modulation systems require different minimum laser linewidths. For example, 4-PSK with a coherent detection function requires a linewidth of ≤100 kHz [15,16,17,18], while currently in 40 Gbps optical communications, the highest-sensitivity high-order 8-QAM, 16-QAM, and 64-QAM signals require a minimum linewidth of 1.2 kHz [19,20]. Linewidth and noise characteristics are important performance indicators for measuring narrow-linewidth lasers. The linewidth restricts the information transmission capacity and optical detection sensitivity in laser communications, while noise affects the transmission quality of laser communications. The applications of narrow-linewidth lasers have become increasingly diverse after decades of development, and the requirements for linewidth characterization technology have become increasingly stringent. The continued development of narrow-linewidth lasers has enabled the linewidth to advance from MHz level to kHz or even Hz level [21,22] while also promoting the advancement of linewidth characterization methods.

The manuscript introduces typical linewidth characterization methods for narrow-linewidth lasers. The characterization methods are classified into two categories based on the principle of linewidth measurement: the beat frequency power spectrum and the phase noise spectrum. The two methods use Voigt contour fitting and phase noise analysis, and the *β* isolation line equation characterizes the laser linewidth. The limiting factors and error sources of the measurement methods are discussed and combined with the current development status of linewidth characterization technology based on the performance parameters and testing conditions of different lasers. In response to the technical difficulties of phase noise jitter and spectrum broadening caused by 1/*f* noise introduced by long-fiber self-heterodynes, this manuscript discusses the iterative algorithm and self-coherent detection method based on short-delay fiber coherent envelope spectra, which improves the range and accuracy of linewidth measurements. Finally, future research directions for linewidth measurement technology are discussed and summarized.

## 2. Basic Principles

The linewidth of a laser mainly derives from random phase jitter, and the recombination and generation of laser carriers and photons can cause instantaneous changes in photon density and carriers. An instantaneous change in photon density leads to amplitude fluctuations in the output power, resulting in background noise. Relaxation oscillation can cause instantaneous changes in carrier density, alter the refractive index, and produce phase changes, resulting in a certain spectral linewidth in the laser mode.

Laser linewidth measurements have typically used spectrometers and F-P standard instruments, with the two measuring linewidth limits in the GHz and MHz orders [23,24], respectively. They cannot meet the linewidth characterization requirements of existing narrow-linewidth lasers. There are several methods for characterizing the linewidths of narrow-linewidth lasers present, including the heterodyne, delayed self-homodyne, delayed self-heterodyne, and noise analysis-based linewidth characterization methods. The experimental setup is shown in Figures 1 and 5.

### 2.1. Heterodyne Beat Frequency Method

The heterodyne method is known as the beat frequency method, which uses two lasers to determine the beat frequency. The wavelength difference between the two must be stable and continuously adjustable to ensure a frequency sweep within a certain frequency range. The linewidth of the reference light source is close to or negligible to that of the measured light source, as shown in Figure 1a. Two laser beams are mixed to generate interference light signals. The beat frequency power spectrum is collected using an electrical spectrum analyzer (ESA), and the measured laser linewidth is calculated based on the beat frequency bandwidth [25,26]. The optical field function is:
(1)$$E_{1}\left(t\right)=E_{1}exp\left\{j\left[2{\pi f}_{1}t+\varphi_{1}\right]\right\}$$
(2)$$E_{2}\left(t\right)=E_{2}exp\left\{j\left[2{\pi f}_{2}t+\varphi_{2}\right]\right\}$$
where *E*_1_ and *E*_2_ are the optical field intensities of the test laser and the reference laser, respectively, *φ*_1_ and *φ*_2_ are the initial phases of the laser, and *f*_1_ and *f*_2_ are the output light frequencies. After coupling two beams of light through a coupler, the beat frequency is detected, and a beat frequency signal is generated, which outputs the corresponding photocurrent. The optical beat current function is:(3)$$I(t)=\beta E_{1}E_{2}cos\left(2\pi △f+△\varphi \right)$$
where *β* is the ratio of the intensity of the two beams, Δ*f* = *f*_2_ − *f*_1_ is the difference in the frequency signal between the two beams, Δ*φ*(*t*) = *φ*_2_ − *φ*_1_ is the phase difference in the dual-frequency light waves, and the self-correlation function of the beat current can be obtained by Fourier transform to obtain the photoelectric current spectral density function:(4)$$S_{b}=△v/2\pi \left[{\left(v-v_{b}\right)}^{2}+{\left(△v/2\right)}^{2}\right]$$
where Δ*v* is the beat frequency linewidth. When the reference linewidth *v_r_* approaches the measured linewidth *v_t_*, the measured linewidth is half of the beat frequency linewidth Δ*v*, as shown in Figure 2a. When the reference linewidth *v_r_* is much smaller than the measured linewidth *v_t_*, the beat frequency linewidth Δ*v* will be approximately equal to the measured linewidth, as shown in Figure 2b.

### 2.2. Delayed Self-Homodyne Method

The delayed self-homodyne method [27] requires only one tested laser, avoiding strict performance requirements for the reference source. The basic structure of the Mach–Zehnder interferometer (MZI) is shown in Figure 1b. First, the tested laser is split by a coupler, and one beam is propagated through the fiber to eliminate coherence and interferes with the other beam to beat the frequency. The mixed-frequency differential beat optical signal is converted into an electrical signal by a photodetector (PD), and the power spectrum of the current signal is analyzed by an ESA [28].

The delayed self-homodyne system converts the phase fluctuation of the measured laser into intensity fluctuation, and the optical signal electric field output by the laser is:(5)$$E\left(t\right)=E_{0}exp\left\{j\left[2\pi f_{0}t+\varphi_{0}\left(t\right)\right]\right\}$$
where *E*_0_ is the intensity of the light field, *f*_0_ and *φ*_0_ are the output frequency and initial phase, respectively. The light intensity electric field of the beat frequency signal is:(6)$$E_{T}\left(t\right)=E\left(t\right)+E\left(t+\tau_{0}\right)$$
where *τ*_0_ is the delay time. The self-correlation function of the light intensity electric field is:(7)$$R_{I}\left(\tau \right)=\langle E_{T}\left(t\right){E_{T}}^{*}\left(t\right)E_{T}\left(t+\tau_{0}\right){E_{T}}^{*}\left(t+\tau_{0}\right)\rangle$$

The simplified PSD function is obtained according to the Wiener–Khinchin theorem:(8)$$S_{T}\left(w\right)=I_{0}\left[v^{2}+\left(2/\tau_{c}\right)/{\left(2/\tau_{c}\right)}^{2}\right]$$
where *τ_c_* is the coherence time. The beat frequency linewidth is twice the linewidth of the laser to be measured, and the measured linewidth is Δ*v* = 1/*τ_c_*.

### 2.3. Delayed Self-Heterodyne Method

The delayed self-heterodyne method involves the addition of an acousto-optic frequency shifter (AOM) based on a self-homodyne, which moves the center frequency to the high-frequency region to avoid interference from the environmental noise near the zero frequency in the system and improves the testing accuracy [29]. The basic structure of the delayed self-heterodyne measurement method includes a MZI, a Michelson interferometer (MI), and a cyclic gain compensation structure.

#### 2.3.1. MZI-Type Delayed Self-Heterodyne Method

The structure of the MZI-type delayed self-heterodyne method is shown in Figure 1c. Its basic principle is the same as that of the aforementioned delayed self-homodyne method. In 1980, T. Okoshi et al. demonstrated a method for characterizing the linewidth of DFB semiconductor lasers [30], which used a delay fiber instead of an intrinsic oscillator. After the tested laser is split into beams, it undergoes a delay fiber delay and an AOM frequency shift. The mixed-beat signal of the delayed light and the frequency-shifted light is:(9)$$I\left(t\right)=E_{L}\left(t\right){E_{L}}^{*}\left(t\right)=E_{L}cos\left[-f_{0}t+\left(f+f_{0}\right)\tau_{0}+\varphi \left(t\right)-\varphi \left(t-\tau_{0}\right)\right]$$
where *E_L_* is the electric field of the interference beat frequency signal. The current power spectral density obtained by Fourier transform of its differential electrical signal is:(10)$$S_{L}\left(f,\;\tau_{0}\right)=\frac{2\left(f\pm f_{0}\right)\tau_{c}}{1+\tau_{c}^{2}{\left(f\pm f_{0}\right)}^{2}}\left\{1-\exp\left(-\frac{\tau_{0}}{\tau_{c}}\right)\left[cos\left(f\pm f_{0}\right)\tau_{0}+\frac{sin\left(f\pm f_{0}\right)\tau_{0}}{\left(f\pm f_{0}\right)\tau_{c}}\right]\right\}+\exp\left(-\frac{\tau_{0}}{\tau_{c}}\right)\delta \left(f\pm f_{0}\right)$$

When the length of the delay fiber is much longer than the coherence length of the laser, *τ*_0_ is much greater than *τ_c_*. The exp term on the right side of the equation approaches 0, and the beat spectrum exhibits a Lorentz shape. The linewidths of the frequency-shifted light and delay light are the same, and the 3 dB bandwidth of the power spectrum curve is twice the linewidth of the tested laser [31,32,33].

Additional optical transmission losses and 1/*f* noise are inevitable for excessively long delay optical fibers, resulting in spectrum broadening and phase noise jitter. Therefore, researchers have proposed MI-type structures [34,35,36,37] and gain-compensated cyclic structures [38,39] based on MZI-type structures.

#### 2.3.2. MI-Type Delayed Self-Heterodyne Method

The structure of the MI-type delayed self-heterodyne method is shown in Figure 1d. The tested laser passing through a 2 × 2 coupler is divided into two paths: delay and frequency shift. A Faraday rotator mirror (FRM) is introduced into the two paths, and the two paths are reflected. Interference superposition is performed on a 2 × 2 coupler [40,41,42]. FRMs can compensate for any rotation of the polarization eigenstates of both arms, reduce noise caused by random polarization state drift, and achieve a delayed optical path twice the length of the fiber. AOM shifts the intermediate frequency twice to 2*f*_0_, away from the zero frequency. The theoretical analysis of its power spectral density and linewidth is similar to that of an MZI-type delayed self-heterodyne and will not be performed.

#### 2.3.3. Gain Compensation Loop Delay Self-Heterodyne Method

The structure of gain compensation loop delay self-heterodyne measurement technology is shown in Figure 1e, which constructs the delay and frequency shift optical path in the MZI delayed self-heterodyne system into a fiber ring structure. The fiber optic ring mainly consists of couplers, AOMs, erbium-doped optical fiber amplifiers (EDFAs), and delay fibers [43,44]. The beam circulates in the fiber-optic ring according to a certain splitting ratio and undergoes an AOM frequency shift once per cycle. The delay of the light also increases with the number of cycles. The EDFA is used to compensate for losses caused by optical field cycling [45,46].

Dawson et al. first proposed the cyclic delay self-heterodyne interferometer [47], while Han et al. conducted a theoretical analysis [48]. Its basic principle is similar to that of the MZI delayed self-heterodyne system, but the difference is that the delayed beam of the cyclic gain compensation system is a set of light waves with a center frequency and delay time in an equal sequence. Beat frequency curves with several peaks can be obtained by using a spectrograph, achieving a high linewidth resolution.

The electrical signal converted from the output light of the system by the photodetector is the nth-order complete current:(11)$$I_{out}\left(t\right)=I_{0}\left(t\right)+\sum_{n=1}^{\infty }I_{n}\left(t\right)$$
where *I_out_* is the DC output component, n is the order of the spectral line, *I_n_* is the nth-order optical direct current, and *I*_0_ is the beat frequency current output with a center frequency of *nω*_0_, which is expressed as:(12)$$I_{0}\left(t\right)=E^{2}{\left(\alpha /2\right)}^{2}cos\left[nf_{AOM}t-n{wf}_{0}\tau_{0}+\varphi \left({t-n\tau }_{0}\right)+\varphi \left(t\right)\right]$$
where *α* is the transmission coefficient of the delay fiber and AOM, and *f_AOM_* is the frequency shift of the AOM. The delay time is much longer than the coherence time of the laser, and the normalized n-order beat current power spectrum is:(13)$$S_{n}\left(w\right)={\left(\alpha /2\right)}^{n}\left(△f/\pi \right)\left[{△f}^{2}+{\left(w-nf_{AOM}\right)}^{2}\right]$$

The loop output only changes the power spectrum gain and does not change the power spectrum type. The effective order of the beat frequency peak is selected, and the average bandwidth is taken. The width of the beat frequency spectrum line is twice the linewidth of the laser to be measured. The effective order delay time should be greater than six times the coherence time to eliminate the influence of the delta pulse function, and the corresponding output power should be able to maintain a complete beat frequency curve. The gain compensation loop delay self-heterodyne has extremely high linewidth resolution but requires multiple cycles within the fiber-optic loop, which is extremely sensitive to the testing system parameters and environmental noise.

#### 2.3.4. Beat Frequency Power Spectrum Analysis

Regarding the impact of 1/*f* noise on linewidth measurements, M. Chen et al. [49,50] proposed a Voigt fitting method to separate the Lorentz components in the beat frequency power spectrum. The beat frequency power spectrum based on the self-heterodyne method is the Voigt spectrum curve, which is a convolution of the Lorentz spectrum and the approximate Gaussian spectrum, as shown in Figure 3. Two types of spectral lines are related to white noise and 1/*f* noise, while 1/*f* noise can cause Gaussian broadening of the spectral lines [51]. A Voigt contour can be expressed as [52]:(14)$$V(v_{v})=\int_{-\infty }^{+\infty }G(v_{v})L(v_{v}-v_{G})dv$$
where *ν_v_* is the frequency of the Voigt spectrum, *ν_G_* is the Gaussian spectral frequency, *G*(*ν*) is the normalized Gaussian line shape with a center of *ν* = *ν*_0_, *L*(*ν*) is the normalized Lorentz line shape with a center of *ν* = *ν*_0_, and *ν*_0_ is the frequency corresponding to the peak of the Lorentz spectral line:(15)$$G(v_{v})=\frac{2\sqrt{ln2}}{\sqrt{\pi }\Delta v_{G}}exp\left[-4ln2{\left(v_{v}-v_{0}\right)}^{2}/{\left(\Delta v_{G}\right)}^{2}\right]$$
(16)$$L\left(v_{v}\right)=\frac{\Delta v_{L}}{2\pi }\frac{1}{{\left(v_{v}-v_{0}\right)}^{2}+\Delta {v_{L}}^{2}/4}$$
where Δ*ν_G_* is the full width at half maximum (FWHM) of a Gaussian line shape and Δ*ν_L_* is the FWHM of the Lorentz line shape. The relationships between the linewidths of the Voigt spectrum (Δ*v_v_*), Lorentz spectrum, and Gaussian spectrum can be approximated as:
(17)$$\Delta v_{V}=\frac{1}{2}\left[1.0692\Delta v_{L}+\sqrt{0.866639{\left(\Delta v_{L}\right)}^{2}+4{\left(\Delta v_{G}\right)}^{2}}\right]$$

The Gaussian component has a strong influence on the 3 dB spectral width based on the broadening effect of 1/*f* noise, while the 20 dB spectral width is mainly affected by the contribution of the Lorentz component. The Lorentz linewidth can be obtained by dividing the 20 dB spectral width by 2√99. The initial values of the Gaussian and Lorentzian linewidths of 3 dB and 20 dB are substituted into Equations (14)–(17) to obtain the Voigt contour curve, which is compared with the measured spectral lines. A calculated value that equates the 20 dB width of the Voigt spectrum with the measured spectral width through iteration is obtained as a new estimated value. The above process is repeated until the estimated value converges, and this process is used as the Lorentz linewidth.

### 2.4. Second-Order Stokes Wave Based on Brillouin Scattering

The method is based on the principle of inducing second-order Stokes light in Brillouin fiber ring lasers and can also be used for linewidth measurements in other lasers [53,54]. The second-order Stokes light is output as a reference light beat with the pumping test beam, and the spectrum is almost identical to the spectral line shape of the test laser [55,56]. The structure is shown in Figure 1f.

The laser under test enters the resonant cavity of the fiber ring through an isolator, and the pump light propagates counterclockwise within the fiber ring, producing first-order Stokes light that propagates clockwise within the ring. The power of the first-order Stokes light reaches the stimulated Brillouin scattering (SBS) threshold when the pump light power is sufficiently high, producing a second-order Stokes light with the same direction as the pump light. The second-order Stokes light is simultaneously output and mixed with the pump light [57,58]. The linewidth of the first-order Stokes light can be narrowed to the Hz level for a sub-MHz pump light, and the linewidth of the second-order Stokes light can be further narrowed to the sub-Hz level [59,60]. The linewidth of the second-order Stokes light is several orders of magnitude smaller than that of the tested laser, and the FWHM of the beat spectrum is equivalent to the linewidth of the tested laser, as shown in Figure 2b.

The method has a simple experimental setup, does not require a long delay fiber, and has high measurement accuracy and lower linewidth measurement limits. However, the laser to be tested needs to maintain single longitudinal mode operation, and there are extensive requirements for the stability of the testing environment.

### 2.5. Linewidth Calculation Method Based on Frequency Noise

The above method achieves accurate linewidth characterization through measuring the beat frequency power spectrum of the measured signal, but the lower limit and measurement accuracy of linewidth measurement through methods such as the delay self-heterodyne methods are limited by the length of the delay fiber and can only achieve a single linewidth expression [61,62,63]. Phase and intensity disturbances are introduced during the stimulated radiation process due to spontaneous emission, which cannot be eliminated in the active region, and fluctuations in quantum noise restrict the lower limit of the linewidth. The impact of frequency noise and phase noise on the linetype should be considered in some application scenarios, and the distribution of the linewidth at the complete Fourier frequency should be analyzed. The measurement of laser noise is also aimed at characterizing the laser linewidth, including the Lorentz linewidth and integral linewidth. The two linewidths are affected by white noise in the high-frequency region and 1/*f* noise in the low-frequency region of the frequency noise power spectrum.

The frequency noise spectrum can be obtained by the phase noise power spectrum, and the calculation formula between the two is as follows [64,65]:(18)$$S_{\delta v}\left(f\right)\left[{Hz}^{2}/Hz\right]=2f^{2}L\left(f\right)$$
where *S*_Δ*v*_(*f*) is the power spectral density of the frequency noise and *L*(*f*) is the power spectral density of the phase noise [66]. A partially intuitive power spectral density function curve is obtained to characterize the phase noise characteristics of the laser in detail by performing corresponding operations on the output light field of the laser [67,68]. The phase noise is mainly related to the following parameters: the complex amplitude *E*(*t*), the power spectral density function *S*(*f*) of *E*(*t*), the phase difference Δ*φ_τ_*(*t*), the FM noise spectrum *S_F_*(*f*), and the phase difference variance Δ*_φ_*(*τ*)^2^. Their interrelationships are shown in Figure 4, and the derived formulas are as follows:(19)$$\Delta \varphi_{\tau }\left(t\right)=\varphi \left(t\right)-\varphi (t-\tau )$$
(20)$$S_{\Delta \varphi_{\tau }}(f)=4{\left[\frac{sin(\pi t\tau )}{f}\right]}^{2}S_{F}(f)E(t)$$
(21)$${\delta (\tau )}^{2}=\langle \Delta {\varphi_{\tau }\left(t\right)}^{2}\rangle$$
(22)$$L(f)=\langle {\left|ℑ\left[E(t)\right]\right|}^{2}\rangle$$
(23)$${\delta (\tau )}^{2}=4\int_{0}^{\infty }{\left[\frac{sin(\pi t\tau )}{f}\right]}^{2}S_{F}(f)df$$
(24)$$L(f)=ℑ\left\{exp\left[-\frac{{\delta (\tau )}^{2}}{2}\right]\right\}$$
where ℑ[•] is the Fourier transform and <•> is the statistical mean.

The frequency linewidth of the laser can be directly determined by the frequency noise and phase noise. The power spectral density of the frequency noise can be integrated using the *β* isolation line algorithm to evaluate the laser frequency linewidth [69]. The laser linetype can be determined by the Fourier transform of the autocorrelation function of the light field according to the Wiener–Khinchin theorem, which are the Lorentz and Gaussian linetypes. The boundary point between these two types in *S*_Δ*v*_(*f*) is determined by the *β* isolation line:(25)$$S_{\delta v}\left(f\right)=8\ln\left(2\right)f/\left(\pi^{2}\right)$$

In the area where *S*_Δ*v*_(*f*) is higher than the *β* isolation line, the laser power spectrum is Gaussian shaped, and *S*_Δ*v*_(*f*) directly determines the frequency linewidth. The integrated linewidth is obtained by integrating the noise component area *A,* as Figure 5a [65,70]:(26)$$△v={\left(8ln2A\right)}^{\frac{1}{2}},\;A=\int_{\frac{1}{T_{0}}}^{\infty }H\left[S_{\delta v}\left(f\right)-8ln2f/\left(\pi^{2}\right)\right]S_{\delta v}\left(f\right)df$$
where *H*(*X*) is a unit step function that has only two values: 0 (*X* > 0) or 1 (*X* < 0).

When the frequency noise is in the high-frequency stage, *S*_Δ*v*_(*f*) is below the *β* isolation line; the laser power spectrum does not contribute to the linewidth, and the frequency noise power spectrum density can be taken as a constant *h*. The laser linetype mainly exhibits a Lorentz distribution, corresponding to the linewidth Δ*ν_L_*:(27)$${△v}_{L}=\pi h$$

In areas dominated by high-frequency noise, the linewidth is only affected by the spectral density *h* of the frequency noise.

Linewidth characterization methods based on laser phase noise measurements and analysis include the frequency discrimination method and coherent detection method.

#### 2.5.1. Frequency Discrimination Method

Frequency discrimination is a method of directly measuring frequency noise based on a frequency discriminator. The frequency discriminator converts frequency fluctuations into intensity fluctuations and obtains frequency changes by detecting intensity changes [71,72]. Its principle and structure are shown in Figure 5b. Frequency discriminators can be used to measure frequency jitter and linewidth, but they cannot directly generate linewidths, making data interpretation more difficult. In addition, the measurement of the frequency discriminator is assumed to be carried out under the condition of continuous operation of the laser, and its intensity will be constant and only change in its optical frequency or phase. The experimental setup and calibration of frequency discriminators are more complex. There are currently multiple interference configurations available for constructing frequency discriminators, such as the MI interferometer, F-P interferometer, and MZI interferometer. Electronic frequency discriminators in the frequency domain have also been developed to improve stability and resolution.

#### 2.5.2. Optical Coherent Reception Method Based on Delayed Self-Homodyne/Self-Heterodyne

There are self-homodyne/self-heterodyne optical coherent reception methods based on delayed self-homodyne difference linewidth measurements [73]. Compared with the MI-type delayed self-homodyne method mentioned earlier [74,75], the optical coherent receiving method introduces an intradyne coherent receiver (ICR) with a 90° mixer at the position of the beam combiner, as shown in Figure 5c. The function of a coherent receiver is to coherently demodulate the signal light and local oscillator light, recover the I/Q component of the laser differential phase noise, obtain the phase modulation noise power spectrum in a digital signal processing (DSP) system, and calculate the integral linewidth and Lorentz linewidth [76,77,78].

#### 2.5.3. Optical Coherent Reception Method Based on a 120° Interferometer

The optical coherent reception method based on a 120° interferometer utilizes the principle of phase reconstruction to characterize phase/frequency noise [79,80,81]. The tested laser passes through a phase difference interferometer system with a 3 × 3 coupler to measure the differential phase information, as shown in Figure 5d. The measured laser is adjusted for power through a variable optical attenuator and is divided into two parts by a ring-shaped device. One part is directly connected to the data acquisition card as the reference light, and the other part is connected to a 3 × 3 coupler. The light passing through output ports 1 and 2 of the coupler is reflected back to the coupler for mixing interference by the FRMs, and output port 3 is unloaded. The three output terminals are split beam lasers with the same amplitude and a phase difference of 120° from each other. The frequency/phase characteristics that deviate from linear scanning from the linear fitting curve are analyzed after the output data are collected through analog cards [82,83]. The phase noise PSD is obtained by demodulating the phase information [84,85].

The linewidth calculation based on phase/frequency noise measurements does not require excessively long fibers, avoiding Gaussian broadening and additional losses. The Lorentz linewidth and integral linewidth calculations are based on the β isolation line algorithm, which is suitable for any type of noise in the tested laser and does not require subsequent fitting of the beat frequency curve. The linewidth is analyzed and calculated by measuring the frequency noise power spectrum. Considering the main contributions of 1/*f* noise and white noise to the laser linewidth, the contributions of phase/frequency noise to the laser linewidth can be observed, and the complete distribution characteristics of noise and linewidth in the Fourier frequency domain can be obtained.

### 2.6. Comparison of Linewidth Characterization Methods

The above linewidth characterization methods were compared and analyzed, and the methods were classified according to their basic principles, as shown in Table 1. The devices and calculation processes of various methods were introduced, and the advantages and disadvantages of each method were analyzed, as well as the limiting factors of the measurement range and accuracy. The characterization methods applicable to different testing environments are listed.

## 3. Research Progress

Guided by the basic principles of the aforementioned linewidth characterization methods, researchers have constructed different types of testing systems and conducted measurements and calculations on laser linewidths of different magnitudes. A systematic study was conducted on the system noise and beat frequency power spectrum fitting that constrain linewidth characterization.

### 3.1. The Heterodyne Method

In 2013, H. Lee et al. proposed a spiral resonator for on-chip laser frequency stabilization and used a Pound–Drever–Hall (PDH) locking system to lock two fiber lasers onto the spiral resonator [86]. The locked laser measures the beat frequency signal through a heterodyne linewidth, achieving linewidth measurements less than 100 Hz.

In 2018, N. Pavlov et al. proposed a narrow-linewidth continuous-wave laser with self-injection locking onto a whispering corridor (WGM) resonant cavity and used a narrow-linewidth tunable fiber laser as a reference laser to construct a heterodyne linewidth measurement system [87], obtaining a Lorentz linewidth of 340 Hz and a Gaussian linewidth of 1.7 kHz.

The heterodyne method offers higher resolution and sensitivity than spectrometers and F-P etalon but is limited by the strict performance requirements of reference light sources, resulting in complex testing systems and higher costs.

### 3.2. Delayed Self-Homodyne Method

In 1998, H. Ludvigsen et al. reported a modified self-homodyne setup with a phase modulator and electrical upconversion [31], which amplifies the photocurrent signal at the detection end using a low-noise amplifier and mixes it with a 200 MHz intrinsic oscillator in a dual balanced mixer. A 460 kHz linewidth measurement was achieved using only 71.7 m of short-delay optical fiber. This structure is rarely used for direct linewidth measurements due to system stability limitations. It is often used for linewidth measurement of optical frequency discriminator (OFD) structures, calculating linewidth based on noise spectra.

In 2019, S. Gundavarapu et al. measured the output of a narrow-linewidth Brillouin laser integrated on a Si_3_N_4_ waveguide platform using a delayed self-homodyne structure [88], achieving an ultra-narrow-linewidth measurement of 0.7 Hz.

In 2021, N. Chauhan et al. constructed a delayed self-homodyne device in the visible light band for linewidth measurement with a 269 Hz visible-light photon integrated Brillouin laser [89].

The method avoids the stringent performance requirements of the reference light source compared with the heterodyne method. However, environmental noise from factors such as temperature changes and air vibrations near the zero-frequency limits the stability and resolution of the system, and the system has strict requirements for environmental stability.

### 3.3. Delayed Self-Heterodyne Method

#### 3.3.1. MZI-Type Delayed Self-Heterodyne Method

In 2013, Y. Peng reported a delayed self-heterodyne method for 100 Hz linewidth measurements [90]. The device is shown in Figure 6a, with a delay fiber length of 10 km to ensure that the delay time is much shorter than the laser coherence time. Narrow-linewidth measurements below 100 Hz can be achieved by fitting the Lorentz curve with bandwidths of −20 dB or −40 dB.

In 2015, K. Kojima et al. proposed a method of separating the intrinsic linewidth and 1/*f* noise of lasers using multiple fiber delayed self-heterodyne methods [91]. The spectral bandwidths corresponding to different intrinsic linewidth and 1/*f* noise are extracted, and the mean square error (MSE) is calculated for each combination. The best fit of the intrinsic linewidth and 1/*f* noise can be obtained from the global minimum value of the surface by least squares fitting. The method can eliminate the 1/*f* noise effect and obtain a much smaller linewidth than traditional measurement methods.

In 2023, J. Chen et al. improved the linewidth measurement system based on the traditional MZI delayed self-heterodyne method [92], as shown in Figure 6b. The half of the delay fiber required for the same optical path is reduced by introducing a ringer and FRM. The polarization states of the two combined beams of light are perpendicular to each other, reducing the influence of thermal and mechanical disturbances on the polarization state in the fiber. The system achieves a linewidth of 2.58 kHz for an 852 nm narrow-linewidth external cavity diode laser (ECDL).

**Figure 6 sensors-24-03656-f006:**
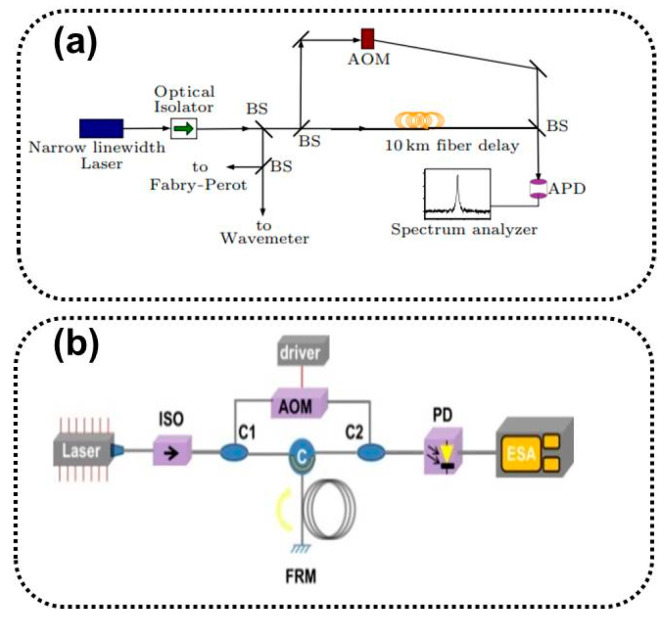
(**a**) Diagram of the self-heterodyne measurement for a narrow-linewidth laser [90]. BS: beam splitter, AOM: acousto-optic modulator, APD: Si avalanche photodiode, FP: Fabry-Perot interferometer. (**b**) Schematic of the laser linewidth measurement [92].

The delayed self-heterodyne method can avoid the zero-frequency noise of the self-homodyne method and improve testing stability. However, the narrower the linewidth is, the longer the delay required for fiber decoherence. Excessive fiber length will result in additional losses, and 1/*f* noise will cause spectrum broadening and phase noise jitter, which limits the detection accuracy and lower limit of linewidth characterization. Therefore, researchers have explored the use of short-delay optical fibers to construct a delayed self-heterodyne linewidth measurement.

In 2016, T. Zhu et al. proposed a coherent envelope amplitude difference comparison method based on short-delay self-heterodyne interferometry for laser linewidth measurements at 100 Hz–1 kHz [93]. The contrast difference between the second peak and the second valley in the strongly coherent envelope (CDSPST) (ΔS) is used to characterize the linewidth, and the system needs to select an appropriate length of optical fiber to maintain ΔS between 10 dB and 30 dB. In 2017, the team further systematically studied a method for accurately measuring ultra-narrow-linewidth based on a high-coherence envelope [94]. The laser linewidth was calculated by analyzing the relationship between the intrinsic linewidth of the laser and the second peak/valley difference in the coherent envelope spectrum, as well as the length of the delay fiber. The correspondence between the laser linewidth and specific CDSPST combinations is calculated, the power spectrum is demodulated, the actual CDSPST values are modified, and the laser linewidth is obtained. The measured linewidth is 150 Hz, with a resolution of 10 Hz. In 2022, S. Huang et al. reported a linewidth measurement scheme based on delayed self-homodyne difference technology [95]. The second peak–valley difference (SPVD) of the power spectrum is used to fit the laser linewidth, avoiding the multiple calculation processes of iterative algorithms.

In 2020, Z. Wang et al. reported an ultra-narrow-linewidth measurement scheme based on partially coherent light interference and dual-parameter acquisition [96]. The laser linewidth is calculated by extracting the power difference between the first-order envelope (between the maximum and minimum points) and the frequency difference between the zero-order minimum point and the center frequency from the power spectrum. The device uses delay optical fibers of km length to reduce the impact of 1/*f* noise and achieve ultra-narrow-linewidth measurements of 100 Hz.

In 2022, Z. Bai et al. proposed an iterative algorithm to demodulate coherent envelope spectra for short-delay optical-fiber self-heterodyne devices [97], which can analyze the beat frequency power spectrum linetype and linewidth. The method obtained the same Lorentz line shape as the long-fiber self-heterodyne method. In the same year, the team proposed a short-delay coherent envelope spectrum demodulation scheme to address the issue of computational speed in iterative algorithms [98]. The relationship between the amplitude difference of a pair of extreme points (maximum and minimum) in the envelope spectrum and the linewidth and fiber length was calculated. The characteristics of coherent envelopes were simulated and experimentally studied, and the influence of background noise on short-delay self-heterodyne/self-homodyne difference linewidth measurements was analyzed [99].

The short-delay fiber self-heterodyne method can solve the problems of the 1/*f* noise broadening spectrum and phase noise jitter caused by fiber length, but there are also shortcomings. On the one hand, it is necessary to estimate the linewidth of the tested laser to select the appropriate length of fiber. On the other hand, the conversion between the power spectrum amplitude difference and the linewidth is not a constant numerical relationship but is related to the setting of the fiber length and needs to be continuously adjusted to avoid the influence of background noise.

#### 3.3.2. MI Delayed Self-Heterodyne Method

In 1995, L. A. Ferreir et al. used an MI delayed self-heterodyne to increase the cumulative testing time of optical frequency fluctuations and suppress interference in the testing environment [41].

In 2011, A. Canagasabey et al. characterized the linewidth of DFB fiber lasers using an MI delayed self-heterodyne [42] and extracted it through Voigt fitting, comparing the Lorentz and Gaussian linewidths and related noise sources. The MI structure has achieved a slightly narrower linewidth than the MZI structure, significantly reducing errors due to the limitation of polarization drift by the FRM.

#### 3.3.3. Gain Compensation Loop Delayed Self-Heterodyne Method

In 1990, H. Tsuchida et al. proposed a gain-compensated cyclic delayed self-heterodyne (R-DSH) method [38]. Multiple transmission fibers and AOM form a loop, increasing the delay time between two laser beams and reducing the fiber length. A PD can obtain infinite orders of photocurrent spectral lines, and their current spectral density can be analyzed through a spectrum analyzer to obtain the linewidth. The technology reduces the cost of fibers and allows for easy measurement of the laser linewidth.

In 2012, H. Tsuchida et al. demonstrated an R-DSH method for high-resolution laser line measurements [45], as shown in Figure 7a. R-DSH increases the delay time through the optical cycle of a heterodyne ring interferometer (HRI). Optical filtering is applied at the HRI output to eliminate and reduce distortion, and the maximum delay of using optical filtering is limited to approximately 180 km. High-order frequency shift components introduce distortion in the beat frequency spectrum, limiting the number of cycles.

In 2021, J. Gao et al. proposed a polarization-insensitive R-DSH method (PI-RDSH) for measuring sub-kHz linewidth [100], as shown in Figure 7b. Introducing FRM effectively reduces polarization-induced attenuation without active polarization control. The combination of the three FRMs ensures that the output polarization state (SOP) of the interferometer is independent of the fiber birefringence, achieving passive polarization-insensitive linewidth characterization.

In 2022, D. Wang et al. demonstrated an improved scheme for a passive R-DSH interferometer for linewidth measurement [101], as shown in Figure 7c. An additional pair of fiber couplers is introduced in traditional R-DSH interferometers to independently balance the power of the reference beam and recursive fiber loop to achieve optimal system performance. The scheme can achieve a greater number of detectable beat frequencies and finer linewidth resolution under the same delay, as shown in Figure 7d.

The beat frequency power spectrum obtained by the cyclic self-heterodyne method is a series of beat frequency signals composed of double-sided bands of laser noise and recirculation oscillation. The equivalent delay time increases with increasing frequency order, and the linewidth of the beat frequency peak tends toward the laser linewidth. Gain compensation reduces the length of the delay fiber and eliminates the influence of 1/*f* noise. The compensation of the gain and the length of the delay fiber need to be repeatedly adjusted to avoid loop oscillation.

#### 3.3.4. Beat Frequency Power Spectrum Analysis

In 2006, X. Chen et al. achieved sub-kHz linewidth measurements using the gain compensated R-DSH method [102]. The signal light circulates multiple times in the delay loop using a fiber amplifier to compensate for the loss of a 25 km delay fiber, resulting in an equivalent delay length of more than a hundred km. The beat frequency power spectrum was fitted using the Voigt function to eliminate spectral broadening caused by 1/*f* noise. The Lorentz linewidth is 0.68 kHz, and it is demonstrated that the measured Lorentz linewidth is independent of the fiber delay length.

In 2015, M. Chen et al. reported an ultra-narrow-linewidth measurement method based on Voigt contour fitting [49], which filtered out the spectral line broadening caused by 1/*f* frequency noise and restored the Lorentz line shape of the laser from the measured self-heterodyne spectrum. The actual spectrum of a single-frequency BEFL with a 45 cm EDF as a mixed-gain medium was measured and compared with that of the direct measurement method, as shown in Figure 8. The fitting linewidth is consistent with that of the heterodyne beat frequency method. The approach improves the resolution and is suitable for any type of laser compared to a long-fiber self-heterodyne.

### 3.4. Second-Order Stokes Light Based on Brillouin Scattering

In 2010, P. Sevillano et al. proposed a linewidth measurement method based on SBS [53], as shown in Figure 9. The principle is to use nonlinear SBS in the low-frequency range to lock the optical frequency of a narrow-linewidth tunable laser source and the measured signal, driving heterodyne mixing. High-power pump light generates first-order Stokes waves with Brillouin inversion and second-order Stokes waves in the same direction. The fluctuation of the optical frequency does not affect the measured beat spectrum, with a resolution of up to 300 Hz, and no longer requires a fiber delay.

In 2019, A. Roy et al. reported and analyzed a linewidth measurement scheme based on stimulated Brillouin-induced self-heterodyne (SISH) technology [103]. The delayed self-heterodyne method achieved a 20 dB linewidth of 15 kHz, which is more effective than the SISH method. However, the SISH method reduced jitter by 10 dB. The SISH method has a lower dependence on the fiber length for linewidth measurement, with a 20 dB linewidth of 23 kHz for a 2 km fiber, which is higher than the measurement value for a 25 km fiber.

In 2023, Z. Bai et al. investigated the effect of pump power density on the Stokes linewidth of different SBS liquid media and achieved controllable linewidth output in the range of 200–400 MHz using FC-43 with a wide Brillouin gain linewidth [104]. The Brillouin gain linewidth and pump power density are the main factors affecting the Stokes pulse linewidth. As the pump power density increases, the Stokes linewidth tends to narrow and approach the pump linewidth. This is the first report to reveal that the narrowing of the Stokes linewidth is limited by the pump linewidth.

Brillouin-induced Stokes light does not rely on delayed fibers compared with other measurement techniques, such as self-interferometry, because the two contributions detected come from the same source and do not require high-frequency modulation such as delayed self-homodyne/self-heterodyne techniques.

### 3.5. Linewidth Testing Method Based on Noise Analysis

#### 3.5.1. Frequency Discrimination Method

In 2019, M. Tran et al. used frequency discrimination to measure frequency noise [105]. The setup is shown in Figure 10a. Using a frequency discriminator, such as a nonequilibrium MZI with a sub-coherent delay difference, the fluctuation in the laser frequency is converted into the output amplitude fluctuation of the interferometer recorded by a photodiode, and the laser frequency power spectral density is obtained. The system is more complex, but it can observe all the noise characteristics of the laser [106]. The use of higher sensitivity frequency discriminators can achieve greater noise suppression.

#### 3.5.2. Optical Coherent Reception Method Based on Delayed Self-Homodyne/Self-Heterodyne

In 2012, T. Huynh et al. proposed a delayed self-heterodyne method based on phase modulation (PM) and a delayed self-homodyne difference method based on a coherent receiver [76] to demonstrate that the in-phase and orthogonal components of the self-heterodyne signal can be demodulated at the first and second harmonics of the phase modulation carrier, while the in-phase and orthogonal components of the self-homodyne difference signal can be demodulated at the orthogonal position of the carrier. The experimental and simulation results were consistent, verifying the feasibility of this technology.

In 2016, E. Conforti et al. used a heterodyne-based offline signal processing method to estimate the laser linewidth and phase noise [82]. The method introduces a heterodyne scheme with data acquisition and offline signal processing, uses time-domain-stored signal analysis data, evaluates coherence time, phase noise, and laser linewidth as functions of bias current, and identifies points where coherent loss occurs to obtain phase noise spectral density and linewidth. The results are provided for lasers with linewidth ranging from 20 kHz to 2 MHz.

In 2017, S. Yamaoka et al. proposed a linewidth measurement scheme based on three-wave interference and digital coherent reception [107]. A three-wave interferometer was used to mix the continuous wave generated by the target laser with the surface acoustic waves of two local oscillators. Complex amplitudes, including three-beat frequencies, are obtained through a dual-channel photodetector, and the data are transmitted to a digital signal processing unit using a digital analog-to-digital converter. The linewidth of each laser can be obtained by solving three simultaneous equations.

In 2018, the team designed a coherent reception method to measure FM noise spectra by mixing three wavelengths with each other [108]. Because the three wavelengths are in pairs without interference, this scheme does not require the use of long fiber delay lines or complex polarization controllers. It can also simultaneously measure the FM noise spectra of the three lasers. The linewidth of the three lasers can be obtained separately based on the three-wave beat frequency signal spectra.

In 2022, Z Yuan et al. reported a correlated self-heterodyne (COSH) method for measuring ultra-low-noise laser linewidth [109] with an experimental setup, as shown in Figure 10b. The three-port AOM combines both frequency shift and beam splitting functions [110,111,112] and can use an AOM with a larger frequency shift range to suppress the RIN. This method can achieve ultra-low-noise laser measurements under a high-frequency offset, with the advantages of low optical power requirements, fast acquisition time, and high-intensity noise suppression.

#### 3.5.3. Optical Coherent Reception Method Based on a 120° Interferometer

In 2015, D. Xu et al. developed a narrow-linewidth laser noise measurement technique based on a 3 × 3 coupler and an unbalanced MI delayed self-heterodyne method [113], as shown in Figure 10c. The system involves injecting the laser under test (LUT) into a 3 × 3 fiber coupler (OC) through a ring resonator. The coupler divides the laser into three paths. The beam passing through ports 1 and 2 is reflected back to the OC through the FRM, while port 3 has no reflection. Three beams of light interfere with the OC, and interference fringes are obtained from ports 1, 2, and 3 on the left. The laser linewidth is obtained by calculating the noise power spectral density.

In 2019, F. Yang et al. used a three-step parallel phase shift measurement method based on a 120° phase difference MI [114] to directly demodulate the accumulated laser differential phase within the delay time, derived the relationship between the differential phase and frequency noise and phase noise, and proposed a dynamic noise characterization method for narrow-linewidth sweep lasers based on phase reconstruction [79].

**Figure 10 sensors-24-03656-f010:**
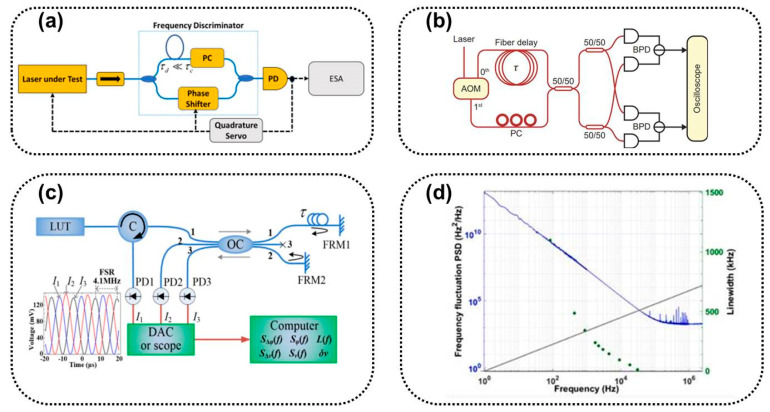
(**a**) Laser frequency noise measurement using a frequency discriminator, which is an unbalanced Mach–Zehnder interferometer [105]; (**b**) COSH setup: A laser is split by a three-port AOM into frequency-shifted (1st-order output) and unshifted (0th-order output) portions [109]; (**c**) 3 × 3 fiber coupler unbalanced Michelson interferometer self-heterodyne technique [113]; (**d**) Measured FN-PSD spectrum with narrowest linewidth [115].

In 2022, X. Luo et al. used a 120° phase difference interferometer to characterize the dynamic noise and differential phase information of a 1550 nm waveguide Bragg grating (WBG) ECDL via phase reconstruction [115]. The phase/frequency noise power spectral density of the laser can be obtained by the β isolation line algorithm, with a minimum integrated linewidth of 4.36 kHz and a minimum Lorentz linewidth of 3.58 kHz, as shown in Figure 10d. In 2023, this team utilized a nonequilibrium MI system to measure the linewidth of an ECDL in 1 μm band [116] and used the β-isolation line integration algorithm to derive and evaluate linewidth in different Fourier frequency domain ranges, resulting in a minimum integrated linewidth of 25.60 kHz and a minimum Lorentz linewidth of 12.00 kHz.

### 3.6. Other Measurement Techniques

There are two commonly used methods for measuring narrow-linewidth lasers based on the beat frequency power spectrum and phase noise analysis. In the process of technological development, these two methods have undergone some improvements, resulting in new linewidth measurement methods.

In 2007, A. Villafranca et al. used a high-resolution spectroscopic analyzer (OSA) to analyze the spectral characteristics of CW emission, which can further characterize the static and dynamic parameters of VCSEL lasers, such as the linewidth, relative intensity noise, and linewidth enhancement factor [117]. The measurement system does not require modulation of the laser, thereby reducing experimental settings to save resources and providing the possibility of measuring all parameters with the same settings.

In 2010, G. Domenico et al. briefly explored and analyzed the relationship between laser frequency noise and laser linearity [118]. A direct and clear evaluation of the relationship between frequency noise and laser linewidth was proposed by analyzing the evolution process of the frequency linewidth of low-pass-filtered white noise lasers changing with the cutoff frequency. The complete characterization of the distribution characteristics can be applied to laser linewidth with any noise spectral density.

In 2013, M. Ravaro et al. reported measuring the intrinsic linewidth of terahertz quantum cascade lasers using near-infrared frequency combs [119]. A beat frequency signal is generated within the radio frequency range based on the repetition frequency doubling of the laser emission frequency and near-infrared laser comb, and a tracking oscillator is used for demodulation. When the output power was 2 mW, the latter was strongly affected by the level of optical feedback, with an intrinsic linewidth of 230 Hz.

In 2016, M. C. Cardilli et al. measured the linewidth of mid-infrared quantum lasers using optical feedback interferometry [120]. The basic principle of the method is to analyze the histogram of the laser’s self-mixing stripe period and directly measure the voltage changes between the laser terminals. The system is shown in Figure 11, where the output beam is collimated by a chalcogenide aspherical lens with an AR coating. A pinhole with a variable diameter is placed in front of the output beam to adjust the feedback intensity and maintain a moderate SM state. Laser self-mixing [121] can avoid external detectors and simplify optical alignment procedures.

We list and compare the linewidth measuring methods with the corresponding device structures and performance parameters for this section and demonstrate the linewidth levels that the method can characterize, as shown in Table 2.

## 4. Comparative Analysis and Technical Difficulties

### 4.1. Comparative Analysis of Linewidth Measurement Techniques

The delayed self-heterodyne is the most widely used and mature linewidth measurement technology, from the MZI structure to the MI structure and cyclic self-heterodyne structure, as well as second-order Stokes waves based on Brillouin scattering. Researchers have gradually optimized the spectrum acquisition, signal analysis, data processing, and testing conditions of the delayed self-heterodyne, effectively improving the measurement range and resolution of linewidth characterization.

The most basic devices for linewidth measurement are optical fibers and photodetectors. The characteristics of the devices and the noise they introduce (such as 1/*f* noise and white noise) significantly restrict the accuracy, range, and precision of linewidth measurements [122,123].

First, the delay time τ_0_ approaches the coherence time *τ_c_* when the length of the delay is insufficient. The exponential function factor in the power spectrum oscillates at both ends of the peak. The beat frequency power spectrum is no longer a standard Lorentz line shape, and a delta function spike appears at the center frequency. The measured signal and reference signal gradually achieve decoherence as the length of the delay fiber increases, and the beat frequency power spectrum approximates the standard Lorentz line shape. The power spectrum will become a Voigt contour line with Gaussian-like components due to the 1/*f* noise of the laser signal itself when τ_0_ is much greater than *τ_c_* [124]. The coherent beat frequency used for linewidth measurement is mainly contributed by the white noise Lorentz line shape, while the 1/*f* noise Gaussian line shape introduces additional spectral line broadening [125]. The frequency spectrum of the external difference signal is corrected to obtain an accurate linewidth by fitting Voigt contour.

Second, linewidth measurements are also limited by devices such as PDs, AOMs and ESAs. PDs introduce thermal noise and shot noise [126,127]; the former is present in almost all transmission media and electronic devices, cannot be eliminated and is not affected by frequency shifts. The latter is related to the emission power and detection bandwidth, which originate from the random fluctuations of charge carriers generated when radiation waves contact a photodetector. The detection bandwidth should be reduced to reduce the impact of shot noise when receiving the complete photoelectric current spectrum. An AOM can lead to issues such as insertion loss and an unstable polarization state, and the complexity of the measurement system increases. We should comprehensively consider the advantages and disadvantages of the AOM and choose whether to shift the frequency for different testing environments. The impact of the spectrum analyzer on the linewidth measurement mainly lies in two aspects: the resolution and the scanning time. All points on the spectrum cannot be accurately calculated simultaneously when the resolution of the spectrum analyzer is too low. The time required to scan the entire RF spectrum is not faster than 1 ms in practical situations, which prevents time dependence research on the linewidth. The linewidth should be measured repeatedly, and the average and standard deviation should be calculated to obtain a more accurate linewidth.

Finally, it is also necessary to consider the impact of the measuring time and the linewidth of the tested/reference light source. The measuring time is reflected in the frequency/phase noise analysis based on the *β* isolation line algorithm, which obtains the linewidth by integrating the power spectral density of the low-frequency modulation region, and the testing time directly determines the lower limit of the integration bandwidth [128,129,130]. It is necessary to consider the linewidth of the measured/reference light source in the heterodyne beat frequency method. One approach is to require the linewidth of the reference light to be approximately equal to the linewidth of the measured light and the linewidth of the beat spectrum to be twice the linewidth of the measured light. Another approach is to ensure that the linewidth of the reference light is much smaller than the linewidth of the measured light, while the spectral linewidth is approximately equal to the linewidth of the measured light [131]. Both of these problems are difficult to satisfy, which limits the use of the heterodyne beat frequency method.

New characterization techniques for narrow-linewidth lasers have emerged in recent years, opening a new chapter for the characterization of narrow linewidth. The second-order Stokes wave based on SBS as a reference light source has become a trending research topic. Its principle has been successfully applied in fiber delay self-heterodyne and F-P interferometers, with a maximum resolution of 300 Hz, achieving sub-Hz-level linewidth characterization. Voigt fitting, the *β* isolation modulation algorithm, and three-wavelength interferometry measurement techniques can be used to calculate the linewidth by measuring the frequency noise [132] and phase noise of lasers, which can be applied to any type of laser. Real-time oscillations of long fibers and lasers can generate 1/*f* noise in delayed self-heterodyne measurement systems. Voigt contour fitting can effectively filter out linewidth broadening caused by 1/*f* noise, while the β isolation modulation algorithm can also separate white noise and 1/*f* noise; therefore, the distribution characteristics of the laser under the influence of noise can be fully observed.

### 4.2. Technical Difficulties and Solutions for Linewidth Characterization

The improvements in characterization methods such as the delayed self-homodyne difference and delayed self-heterodyne have improved the resolution of linewidth measurements. However, the delayed self-homodyne difference beats at zero frequency as the center frequency, which is greatly affected by external factors such as temperature and mechanical oscillation. The delayed self-heterodyne shifts the beat signal frequency by modulating the laser and still relies more commonly on complex measurement systems with long delay fibers. The long length of the fiber introduces 1/*f* noise, leading to broadening of the beat frequency spectrum. Therefore, both methods are susceptible to 1/f noise, which can cause the laser phase noise to shake.

Short delay fibers can lead to coherent envelope distinct beat spectra, and the laser linewidth can be obtained by fitting the beat spectra [133,134,135]. We have successively proposed coherent envelope demodulation methods based on iterative algorithms [136,137] and self-coherent detection methods based on strong coherent envelopes [138].

#### 4.2.1. Coherent Envelope Demodulation Based on an Iterative Algorithm

Coherent envelope demodulation based on an iterative algorithm divides the short-fiber delay self-heterodyne power spectrum into three parts, as shown in Figure 12a: Lorentz line spectrum *S_1_*, periodic modulation function spectrum *S_2_*, and delta function spike spectrum *S_3_*.
(28)$$S\left(f,\Delta f\right)=S_{1}S_{2}+S_{3}$$
(29)$$S_{1}=\frac{P_{0}^{2}}{4\pi }\frac{\Delta f}{\Delta f^{2}+{\left(f-f_{0}\right)}^{2}}$$
(30)$$S_{2}=1-exp(-2\pi \Delta f\tau_{d})\left\{\cos\left[2\pi \left(f-f_{0}\right)\tau_{d}\right]+2\pi \Delta f\tau_{d}sinc\left[2\pi \left(f-f_{0}\right)\tau_{d}\right]\right\}$$
(31)$$S_{3}=\frac{P_{0}^{2}}{2\pi }exp(-2\pi \Delta f\tau_{d})\delta (f-f_{0})$$
where *P*_0_ is the output power of the measured laser, *f* is the frequency of the beat frequency signal, *f*_0_ is the frequency shift, and Δ*f* is the measured laser linewidth. S (*f*, Δ*f*) is the result of the Lorentz linetype spectrum *S*_1_ being modulated by a periodic function *S*_2_, where the period and amplitude of the *S*_2_ function are 1/*τ*_0_ = c/*nL* and exp(−2πΔ*fτ*_0_), respectively; these two parameters are determined by the delay length and the tested linewidth. Using a step-by-step iterative algorithm to estimate the linewidth and achieve coherent envelope demodulation, the process is shown in Figure 13:

The algorithm loops continuously until the estimated linewidth Δ*v_est_* converges. After multiple iterations and constant revisions, Δ*v_est_* is used to obtain more accurate estimates.

The method effectively avoids the effects of spectrum broadening and phase noise fluctuations caused by 1/*f* noise, resulting in more accurate measurement results. However, this algorithm requires multiple iterations and still needs to be improved in terms of computational speed. In addition, the shorter the delay fiber is, the more pronounced the coherent envelope spectrum, as shown in Figure 12b. If the delay fiber is too short, it will cause the “valley” of the coherent envelope spectrum to be submerged by background noise, as shown in Figure 12c, requiring a more sensitive detector and a spectrum analyzer with lower background noise. It is necessary to choose an appropriate range of fiber lengths for linewidth characterization.

#### 4.2.2. Self-Coherent Detection Based on a Strong Coherent Envelope

When the delay time is equivalent to (or shorter than) the coherence time, due to the influence of delta function modulation and AOM frequency shift stability, there is a significant spike in the center frequency of the spectral line, and its amplitude is not conducive to linewidth characterization. We analyze the contrast difference between the second peak and second valley of the envelope spectrum (CDSPST) and the relationship between the linewidth and the length of the delay fiber based on the strong coherent envelope self-coherent detection method. The linewidth is calculated as shown in Figure 12d.

The delay time will be much shorter than the coherence time when using short-delay fiber-optic measurements. At this point, the beat frequency photocurrent power spectrum can be represented as a Lorentz function *S_1_* and a quasiperiodic function *S*_2_. The envelope period of the beat frequency power spectrum is related to *S*_2_, and the period of *S*_2_ depends on the product of the delay fiber length and linewidth. The functions containing delta and Taylor series expansions were applied to the exp term, ignoring high-order terms:(32)$$S\left(f\right)=△v\frac{P^{2}}{4\Pi {\left(f-f_{0}\right)}^{2}}\left\{1-(1-\frac{2\Pi {l_{s}n}_{s}}{c}△v)\left[cos\frac{2\Pi {l_{s}n}_{s}(f-f_{0})}{c}+\frac{sin(f-f_{0})\frac{2\Pi {l_{s}n}_{s}}{c}}{f-f_{0}}\right]\right\}$$
where Δ*v* is the Lorentz linewidth, *f*_0_ is the center frequency, and the frequency difference between the minimum points of each order and the center frequency of the power spectrum Δ*f_m_* can be expressed as:(33)$$△f_{k}=\frac{(k+1)c}{{l_{s}n}_{s}}$$
where *f_AOM_* is the frequency shift of the AOM and *k* is the order of the minimum point. The maximum and minimum points of the power spectrum curve are determined through the maximum and minimum points of *S_2_*:(34)$$\begin{array}{ll} △S\left(△v\right)= & 10{log}_{10}^{S_{peak}}-10{log}_{10}^{S_{trough}}=10{log}_{10}^{\left(S\left(△f-\frac{\left(2l-1\right)c}{2nL},△v\right)/S\left(△f-\frac{mc}{nL},△v\right)\right)}\\ & =10{log}_{10}^{\left(\left[1+{\left(2c/△vLn\right)}^{2}\right]\left[1+exp\left(-2\pi △\frac{vLn}{c}\right)\right]/\left[1+{\left(3c/2△vLn\right)}^{2}\right]\left[1+exp\left(-2\pi △\frac{vLn}{c}\right)\right]\right)(m=2,L=2)} \end{array}$$

The delay length and linewidth have a corresponding relationship. When Δ*S* and *L* are known, the linewidth can be determined. The measurement and calculation method using a coherent envelope can avoid the influence of 1/*f* noise on the measurement range and accuracy of the linewidth and achieve accurate measurement of the linewidth.

## 5. Outlook

The linewidth characterization technology of narrow-linewidth lasers is gradually maturing and can characterize linewidth on the order of 100 Hz. The manuscript proposes several prospects in response to the current situation. First, for the most widely used delayed self-heterodyne technology, the noise caused by double Rayleigh scattering in delay fibers can affect the shape of spectral lines [139,140], leading to center frequency offsets and significant frequency noise, which can introduce inaccuracy into the linewidth measurement. The introduction of 1/*f* noise caused by excessively long delay fibers in the heterodyne method can lead to phase noise jitter and beat frequency spectrum broadening, which is a key issue to consider. Second, the characterization techniques for the linewidth enhancement factor need to be improved [141,142]. One of the key parameters of semiconductor optical amplifiers is the linewidth enhancement factor [143], which is defined as the ratio of changes in the real and imaginary parts of the complex refractive index and is an important parameter affecting the output characteristics of semiconductor lasers. The free-space interference method in the measurement of linewidth factor [144] needs to be carried out on an optical platform, and laser alignment and environmental fluctuations can cause instability factors. The spectrum analyzer in RF modulation measurement methods requires direct use of high-frequency-modulated laser sources [145], and further improvements to corresponding characterization techniques are needed to avoid spatial interference and improve measurement accuracy. Finally, the constraints of environmental noise and measurement system noise on the testing process should be reduced or eliminated [146,147]. The detection range and accuracy of linewidth characterization can be improved by controlling precise and stable experimental testing conditions and using ultra-low-noise linewidth characterization testing devices.

## 6. Conclusions

In this work, we summarize the linewidth characterization techniques of narrow-linewidth lasers. The characterization methods are classified into two categories based on the principle of linewidth measurement: those based on the beat frequency power spectrum and those based on phase noise analysis. The advantages and disadvantages of various testing schemes are analyzed. From the above research progress, it can be concluded that laser linewidth of different magnitudes can be characterized under different experimental conditions. Through comparative analysis, the fiber delay self-heterodyne method is widely used in the measurement schemes of narrow-linewidth lasers, while second-order Stokes light based on Brillouin scattering is used as a reference light source to drive heterodyne mixing, achieving linewidth characterization below the kHz level. In response to the shortcomings of existing research, it is proposed that the focus be on addressing the issues of phase noise jitter and beat frequency spectrum broadening caused by the 1/*f* noise introduced by excessively long delay fibers, as well as the coherent envelope spectrum being submerged by the background noise caused by excessively short delay fibers, to reduce experimental costs and improve measurement accuracy. At the same time, future research trends in linewidth characterization methods are discussed.

## Figures and Tables

**Figure 1 sensors-24-03656-f001:**
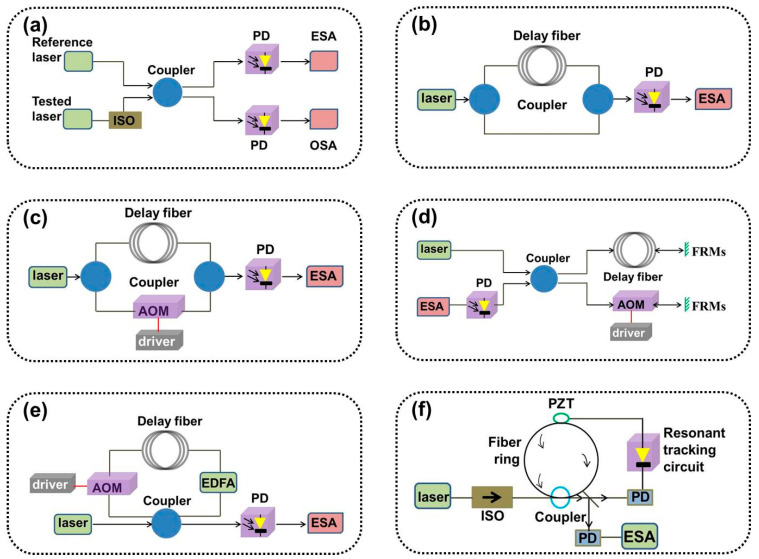
Basic schematic diagram of (**a**) the heterodyne beat frequency method, (**b**) the delayed self-homodyne method, (**c**) the MZI delayed self-heterodyne method, (**d**) the MI delayed self-heterodyne method, (**e**) the cyclic gain compensation delayed self-heterodyne method, and (**f**) the Brillouin second-order Stokes wave. ISO: isolator, PZT: piezoelectric ceramic, PD: photodetector, ESA: spectrum analyzer.

**Figure 2 sensors-24-03656-f002:**
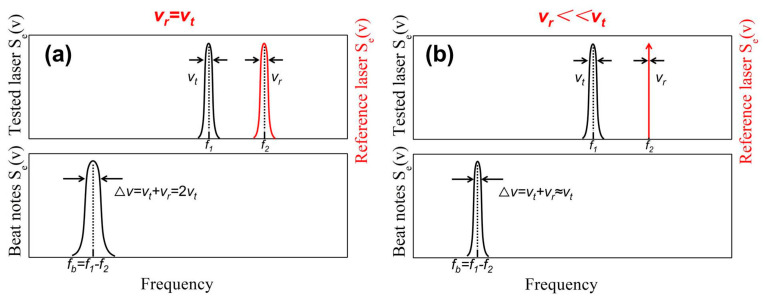
Schematic diagram of the beat frequency signal when the reference linewidth (**a**) is equal to and (**b**) is much smaller than the measured linewidth.

**Figure 3 sensors-24-03656-f003:**
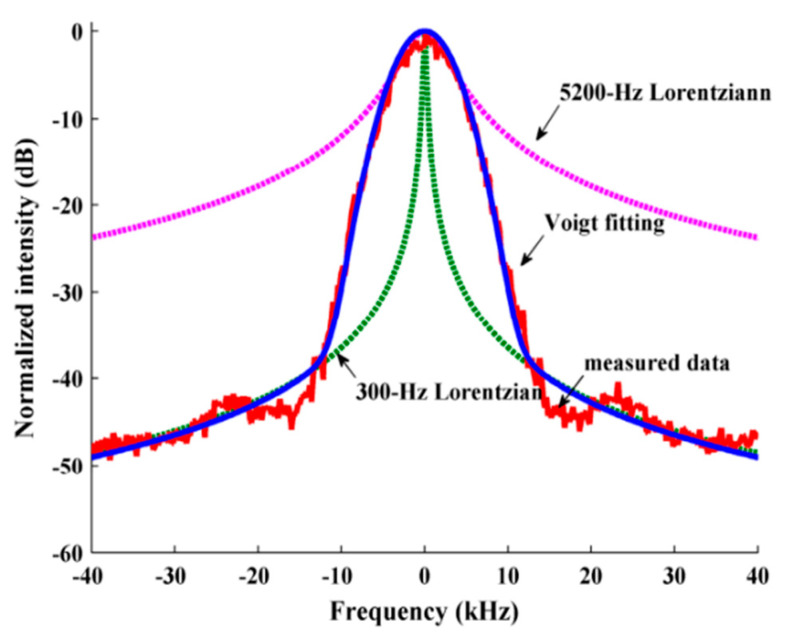
Self−heterodyne spectrum of the Brillouin/erbium fiber laser (BEFL) measured with a 25 km delay fiber and the fitting curves [49].

**Figure 4 sensors-24-03656-f004:**
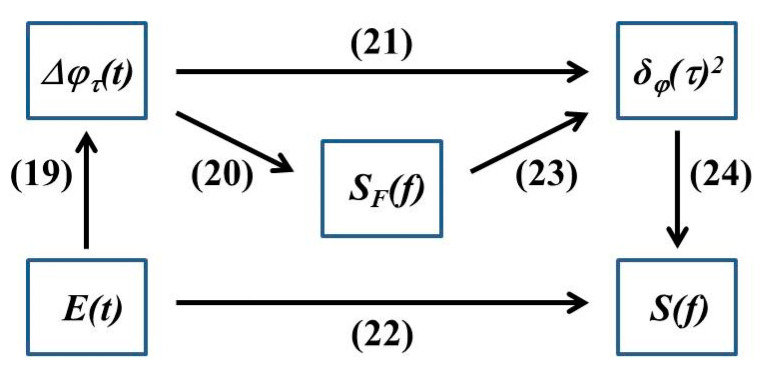
The relationship between commonly used representations of phase noise.

**Figure 5 sensors-24-03656-f005:**
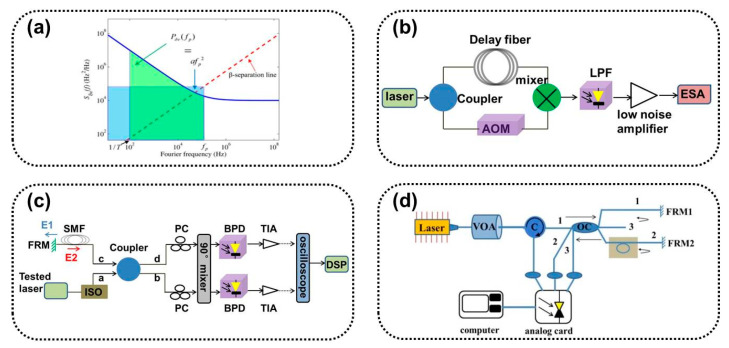
(**a**) Schematic diagram of PSD with frequency noise [65], (**b**) basic structure of the frequency discrimination method, (**c**) self-homodyne optical coherent reception method with FRM, and (**d**) optical coherent reception method based on a 3 × 3 coupler.

**Figure 7 sensors-24-03656-f007:**
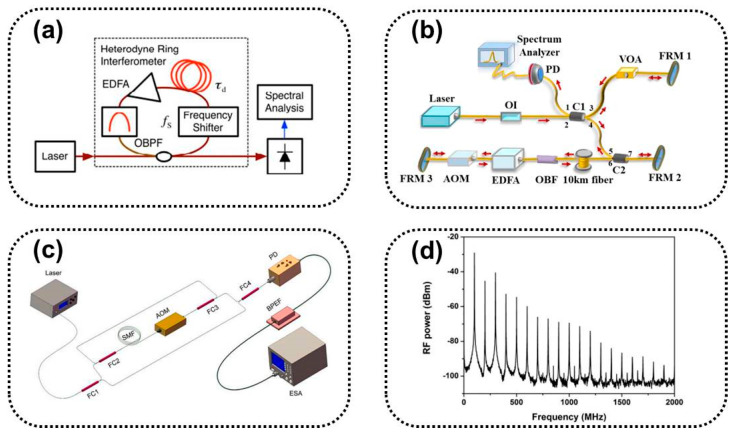
(**a**) Schematic of the laser lineshape measurement apparatus based on the R−DSH method [45], (**b**) schematic of the PI-RDSH laser linewidth measurement setup [100], (**c**) experimental setup of the improved recirculating delayed self-heterodyne interferometer [101], and (**d**) beat signals detected by the proposed RDSHI [101].

**Figure 8 sensors-24-03656-f008:**
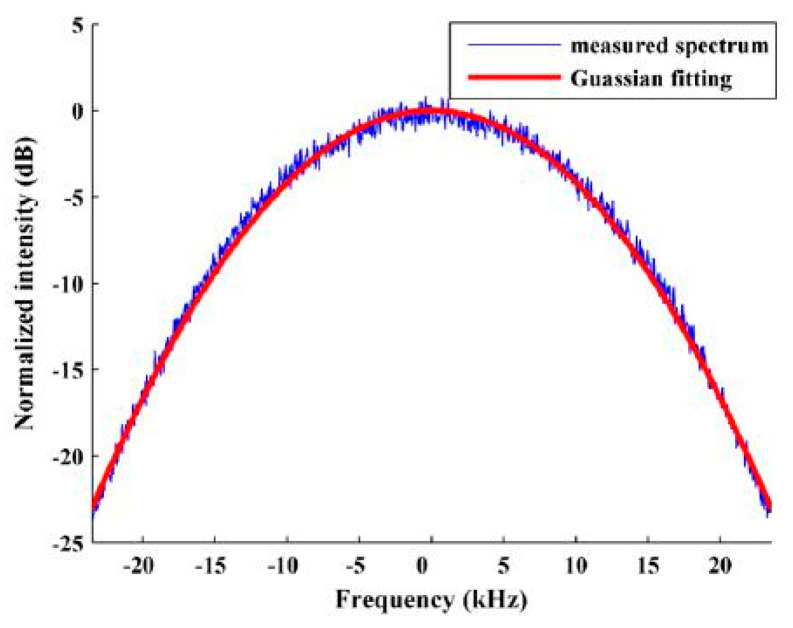
The BEFL self−heterodyne spectrum and its fitting curve were measured using a 100 km delay fiber [49].

**Figure 9 sensors-24-03656-f009:**
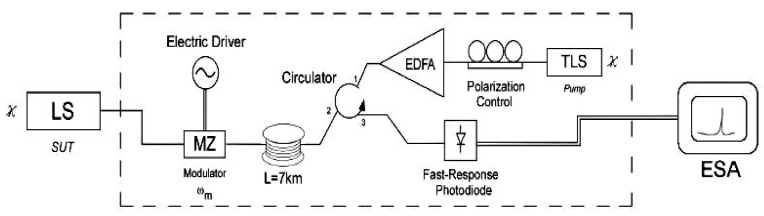
Delayed self-heterodyne measurement system based on SBS [53].

**Figure 11 sensors-24-03656-f011:**
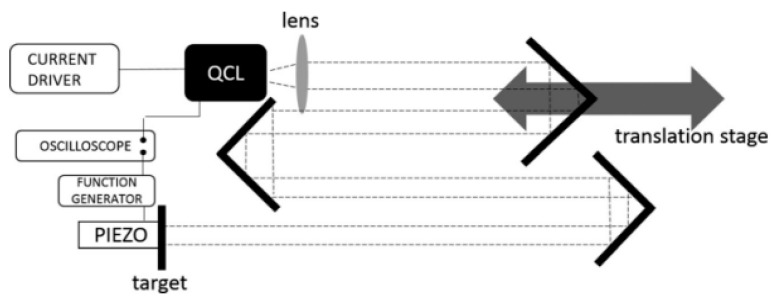
Experimental devices for measuring the mid-infrared QCL linewidth via self-mixing interferometry [120].

**Figure 12 sensors-24-03656-f012:**
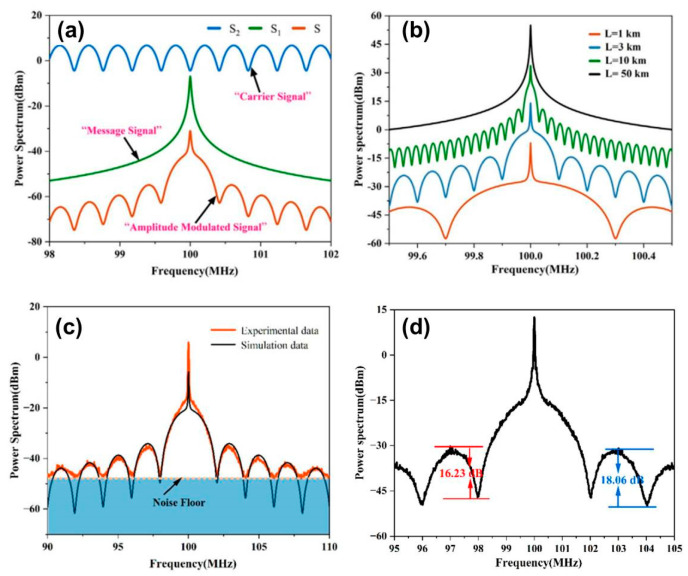
(**a**) Composition of the beat frequency power spectrum [98], (**b**) coherent envelope spectra of different fiber lengths [99], (**c**) schematic diagram of the valley of the envelope spectrum submerged by background noise [99], (**d**) contrast difference between the second peak and the second valley of the envelope spectrum (CDSPST) [98].

**Figure 13 sensors-24-03656-f013:**
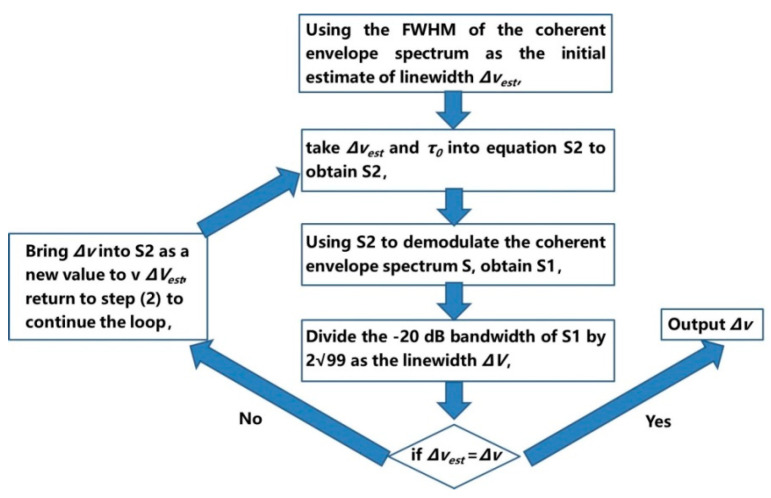
Iterative algorithm flowchart based on coherent envelope spectrum demodulation.

**Table 1 sensors-24-03656-t001:** Comparison of linewidth characterization methods.

Methods	Principles	Advantages	Disadvantages	References
Heterodyne	Beat frequency	—	Strict requirements for reference light sources	[26,27]
Delayed self-homodyne	Mach–Zehnder configuration	Avoiding dependence on reference laser	Zero-frequency signal affect the beat frequency	[28,29]
MZI delayed self-heterodyne	Decoherent coupling	Avoiding interference from zero frequency	Long fiber length can introduce 1/*f* noise	[30,31,32,33]
MI delayed self-heterodyne	Decoherent coupling	Reduce fiber by half for same optical path	Long fiber length can introduce 1/*f* noise	[40]
Loop delayed self-heterodyne	Gain compensation	Can measure wide range of laser bands	Unable to eliminate the impact of 1/*f* noise	[41,42]
Second order Stokes wave	Beat frequency	High measurement accuracy	Sensitive to system parameters	[53,54]
Frequency discrimination	FM-AM	Avoiding interference from zero frequency	Frequency range is limited and precise control is required	[71,72]
Optical coherent reception	Coherent mixing	Directly measure instantaneous phase change	Complex structure and high cost	[73,74,75]

Table note: FM: frequency modulation. AM: amplitude modulation.

**Table 2 sensors-24-03656-t002:** Comparison of other measurement techniques.

Structures	Methods	Linewidth	Output Power	Wavelength	Time
DFB Laser	High resolution spectral analyzer	8.7 MHz	—	1550 nm	2005 [122]
VCSEL laser		—	1.95 mW	1550 nm	2007 [117]
Quantum cascade laser	Near-infrared frequency comb	230 Hz	2 mW	780 nm	2012 [119]
Brillouin fiber laser	Voigt fitting	40 Hz	3 mW	—	2015 [49]
Mid-infrared quantum laser	Optical feedback interferometry	280 kHz	—	—	2016 [120]

## Data Availability

Data underlying the results presented in this paper are not publicly available at this time but may be obtained from the authors upon reasonable request.

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
