# Peer review of "Linewidth Measurement of a Narrow-Linewidth Laser: Principles, Methods, and Systems"

_sensors, 2024, doi:10.3390/s24113656_

Round 1

Reviewer 1 Report

Comments and Suggestions for Authors

 This article is a comprehensive review that systematically introduces the principles of two types of methods for measuring the linewidth of narrow linewidth lasers, which are based on the measurement of beat frequency power spectrum and phase noise analysis. The advantages and disadvantages of each measurement method are also analyzed. Furthermore, the current progress in linewidth measurement is reviewed, and the development direction of the measurement methods for the characteristics of narrow linewidth lasers is discussed. I believe that this manuscript is suitable for publication.

Author Response

Enclosed please find the responses to the comments of the reviewers. All the modifications and supplements are highlighted in red color in the revised manuscript.

Reviewer 2 Report

Comments and Suggestions for Authors

Dear Authors and Editors!

Thank you for the opportunity to evaluate the manuscript. Though I am not exactly a specialist in the field of precise laser measurements, it was really interesting and informative to read the paper, and I would recomment to accept the manuscript. However, the work should be re-read prior to publication to correct minor inaccuracies; the examples here are the arrow diagram in Fig.1d  (2x2 coupler is needed) and the convolution in Fig.3 (Lorentzian -> Gaussian). Once these and similar issues are corrected, the paper may be published in "Sensors".

Comments on the Quality of English Language

In my opinion, English is ok in the manuscript

Author Response

(The authors gave the same response as above.)

Reviewer 3 Report

Comments and Suggestions for Authors

The review is devoted to methods for measuring the linewidth of lasers. It examines a range of different methods and analyzes their advantages and disadvantages. Despite the fairly complete coverage of information on the topic, I have a number of comments.  Firstly, the title does not fully reflect the content of the work. The title includes the term “narrow linewidth laser,” but from the text and pictures it is clear that this is not an arbitrary type of laser, but a fiber one. In addition, the concept of a narrow line is also vague - for laser spectroscopy a narrow line is hundreds of hertz or less, for diode lasers it is on the order of kHz, for metrology the frequency is a unit of hertz. It would be useful to somehow clearly define which cases are considered in the review. There is confusion in the notation: in equation 5 the frequency is designated as f, in equation 8 it is  \nu. There are quite a lot of typos and stylistic errors. For example section 2.3.1 5th line: "it undergoes delay fiber delay..."

I didn't understand why authors didn't mention any advantage of heterodyne method. My opinion this is the most accurate one and the only one that allows you to reliably measure linewidths at the level of units of hertz without performing spectroscopic experiments.

The generally accepted designation for semiconductor lasers with an external cavity is ECDL (external cavity diode laser).

As a rule, the authors of the review have their own contribution to the issues under consideration, but I was unable to highlight it in the text. I would recommend highlighting the authors' contributions, if any. If not, then it is necessary to explain why they decided to do the review and why they are considered experts in this field.

Author Response

(The authors gave the same response as above.)
